# Impact of Processing Methods on the *In Vitro* Protein Digestibility and DIAAS of Various Foods Produced by Millet, Highland Barley and Buckwheat

**DOI:** 10.3390/foods12081714

**Published:** 2023-04-20

**Authors:** Lulu Fu, Song Gao, Bo Li

**Affiliations:** Beijing Laboratory for Food Quality and Safety, College of Food Science and Nutritional Engineering, China Agricultural University, No.17, Qinghua East Road, Haidian District, Beijing 100083, China

**Keywords:** simulated human digestion, thermal processing, cereal-based foods, protein quality, millet, highland barley, buckwheat

## Abstract

Cereals are rich sources of dietary protein, whose nutritional assessments are often performed on raw grains or protein isolates. However, processing and gastrointestinal digestion may affect amino acid (AA) compositions, then change the protein quality. In this study, we determined the digestibility and AA compositions of various foods produced by whole grains (PG) or flour (PF) from three cereals (millet, highland barley and buckwheat) and analyzed the impact of processing methods on the digestible indispensable amino acid score (DIAAS) using the INFOGEST protocol. The *in vitro* protein digestibility of cereal-based foods was lower than raw grains, and PF showed a better digestion property than PG. The intestinal digestibility of individual AA within a food varied widely, and the digestibility of Cys and Ile was the lowest among all AAs. The DIAAS values of PG were lower than those of PF in each kind of cereal, and PF of buckwheat had the highest DIAAS value, followed by highland barley. The first limiting AA was still Lys for millet and highland barley compared to the raw grains; however, for buckwheat it was Leu. This study provided nutritional information on cereal products and helped to guide the collocation of different foods in diets.

## 1. Introduction

Millet is an important food crop of the arid and semi-arid areas in Africa and Asia, dating back 7000 years [1]. Buckwheat is consumed as a staple food in most areas of Asia and North America [2]. Highland barley is mainly cultivated in the Tibetan area, also known as “Qingke” in Chinese [3]. Cereals are eaten as a staple food of the human diet, which supply a majority of the dietary protein for humans, especially in developing countries [4,5].

Thus, accurately assessing the nutritional protein quality of cereal-based foods is necessary. Cereals are usually cooked using grains or flour to make various foods such as rice, steamed buns and noodles. Cooking can affect the nutritional and physicochemical properties of the food proteins as a consequence of AAs’ oxidation, isomerization and reactions with reducing sugars, polyphenols and tannins [6,7,8].

In addition, cereal-based foods have to undergo gastrointestinal (GI) digestion before nutrients absorption. The metabolic availability of an individual AA may vary widely within a food, and the availability of the limiting AA determines the level of all AAs utilized in protein synthesis and other metabolic activities [9].

However, directly determining dietary protein and AA digestibility in humans often poses thorny ethical issues. FAO has recommended dietary AA as an individual nutrient and a utilized digestible indispensable AA score (DIAAS) as an indicator of protein quality, which includes the measurement of ileal digestibility [10]. Nevertheless, these experiments performed on animals are difficult, costly, time-consuming and raise serious ethical problems [11,12]. 

Thus, a harmonized *in vitro* digestion protocol was developed by the COST (European Cooperation in Science and Technology) action INFOGEST (an international network of excellence on the fate of food in the gastrointestinal tract), with the aim of closely mimicking the digestion processes in humans [13,14]. The INFOGEST static protocol was validated for its biological relevance on milk proteins using pigs as an animal model [15], and in comparison with human digests [16]. Furthermore, the INFOGEST static model was validated for the quantitative analysis of protein and individual AA digestibility and the subsequent determination of DIAAS on different protein sources (black beans, wheat bran cereal, pigeon peas, et al.) using pigs as an animal model [17] and using human digests as a comparison [18].

In this study, we determined the protein digestibility and AA compositions of various foods produced by whole grains (PG) or flour (PF) from three cereals (millet, highland barley and buckwheat) and analyzed the impact of processing methods on the digestible amino acid score (DIAAS) using the INFOGEST protocol. This study helped to guide the collocation of different foods in diets to improve the protein and AA availability.

## 2. Materials and Methods

### 2.1. Plant Material

Highland barley (variety of Beiqing 3), millet (variety of Eastlight) and buckwheat (variety of Largetrigon) were purchased from Dongfangliang agri-foods Co., ltd. (Datong, China).

### 2.2. Protein Extraction

Dried grains were ground using a JYL-C91T grinder (Joyoung Co., LTD, Jinan, Shandong, China) with 60 mesh sieves to make wholegrain flour. The flour was mixed with hexane (1:10, *w*/*v*) for 4 h at 37 °C to remove the lipid. The precipitate was collected and air-dried to obtain defatted wholegrain flour.

The defatted millet flour was mixed with deionized water (1:7, *w*/*v*), and three enzymes (amylase, glucoamylase and cellulase) were added at pH 5.0 to hydrolyze starch and cellulose [19]. After centrifugation (9000 rpm for 15 min), the precipitate was collected and freeze-dried as millet protein isolates.

The protein isolates from defatted highland barley flour were prepared by isoelectric precipitation [20]. Briefly, the flour was mixed with deionized water (1:10, *w*/*v*) and stirred continuously for 30 min (pH 9.5, 45 °C). The mixture was then centrifuged for 10 min at 4500 rpm. After centrifugation (4500 rpm for 10 min), the supernatants were collected and adjusted to pH 4.0. The supernatants were removed at the end of centrifugation, followed by washing twice with distilled water. The precipitate was neutralized using 2 M NaOH and freeze-dried as highland barley protein isolates.

The extraction of protein isolates from defatted buckwheat flour followed the protocol by Kayashita et al. [21] with several adaptations. Briefly, the flour was mixed with deionized water (1:10, *w*/*v*) and stirred for 30 min (pH 8.0, 40 °C). After centrifugation, the supernatants were collected and adjusted to pH 4.5. The precipitate was collected at the end of centrifugation and washed twice with distilled water. The precipitate was neutralized to pH 7.0 using 2 M NaOH and freeze-dried as buckwheat protein isolates.

### 2.3. Food Processing

PG meant cooked grains prepared using a commercially available cooker (passing 100 °C steam), according to the method established by Liu et al. [22] with some modifications. For millet, the grains were soaked for 30 min at 25 °C and cooked in water (1:1.5, *w*/*v*) for 25 min. For highland barley, the grains were soaked for 30 min at 25 °C and cooked in water (1:2, *w*/*v*) for 35 min. For buckwheat, the grains were soaked for 30 min at 25 °C and cooked in water (1:2, *w*/*v*) for 30 min.

PF included millet steamed buns, highland barley noodles and buckwheat noodles. Dried grains were ground using a JYL-C91T grinder (Joyoung Co., LTD, Jinan, Shandong, China) with 60 mesh sieves to make flour. The preparation of millet steamed buns was as follows: For each 100 g of millet flour, 48 mL of deionized water and 1.0 g of yeast were added. The mixture was kneaded manually to make dough and allowed to rest for 20 min at 38 °C. Then the dough was kneaded manually into a ball and steamed for 20 min at 100 °C. The preparation of highland barley noodles was as follows: For each 100 g of highland barley flour, 2.0 g of salt and 70 mL of distilled water were added. The mixture was kneaded manually to make dough and allowed to rest for 20 min at 38 °C. After this, the dough was sheeted and cut into strands of 2.0 mm width and 2.0 mm thickness and steamed for 17 min at 100 °C, which was named highland barley noodles. The preparation of buckwheat noodles basically coincided with highland barley noodles but varied in the addition of water (for each 100 g of buckwheat flour, 55 mL of distilled water was added).

After processing, all cereal-based foods were cooled down to 40 °C and subjected to *in vitro* digestion.

### 2.4. In Vitro Digestion

*In vitro* digestion methods were performed according to the INFOGEST static protocol [13], with some modifications for bile concentration according to the protocols by Zhang et al. [23] and Ding et al. [24].

#### 2.4.1. Preparation of Simulated Digestion Fluids

The electrolyte stock solutions of simulated digestion fluids included simulated salivary fluid (SSF), simulated gastric fluid (SGF) and simulated intestinal fluid (SIF).

(i) The preparation of the electrolyte stock solution of SSF was as follows: 15.1 mL of 0.5 M KCl, 3.7 mL of 0.5 M KH_2_PO_4_, 6.8 mL of 1M NaHCO_3_, 0.5 mL of 0.15 M MgCl_2_(H_2_O)_6_ and 0.06 mL of 0.5 M (NH_4_)_2_CO_3_ were mixed and diluted with ultrapure water to 400 mL. The pH was adjusted to 7.0 with 6 M HCl;

(ii) The preparation of the electrolyte stock solution of SGF was as follows: 6.9 mL of 0.5 M KCl, 0.9 mL of 0.5 M KH_2_PO_4_, 12.5 mL of 1M NaHCO_3_, 11.8 mL of 2 M NaCl, 0.4 mL of 0.15 M MgCl_2_(H2O)_6_ and 0.5 mL of 0.5 M (NH4)_2_CO_3_ were mixed and diluted with ultrapure water to 400 mL. The pH was adjusted to 3.0 with 6 M HCl;

(iii) The preparation of the electrolyte stock solution of SIF was as follows: 6.8 mL of 0.5 M KCl, 0.8 mL of 0.5 M KH_2_PO_4_, 42.5 mL of 1M NaHCO_3_, 9.6 mL of 2 M NaCl and 1.1 mL of 0.15 M MgCl_2_(H_2_O)_6_ were mixed and diluted with ultrapure water to 400 mL. The pH was adjusted to 7.0 with 1 M NaOH.

#### 2.4.2. *In Vitro* Digestion

*In vitro* digestion comprised simulated oral digestion, simulated gastric digestion and simulated intestinal digestion. The enzyme activities were measured according to the INFOGEST static protocol [13], with an amylase activity of 13 U/mg, pepsin activity of 500 U/mg and trypsin activity in pancreatin of 6.5 U/mg. The entire digestion was performed in 50 mL centrifuge tubes, shaken at 37 °C and 100 rpm. Control experiments were carried out with ultrapure water.

Simulated oral digestion: Each sample was minced by a JYL-C93T electric mincer (Joyoung Co., LTD, Jinan, Shandong, China). The substrate solution was mixed with SSF at 1:1 (*v*/*v*) ratio, containing 75 U/mL α-Amylase and 1.5 mM CaCl_2_. The procedure was as follows: 5 mL of the substrate solution (0.15 g protein) was mixed with 3.5 mL of the SSF electrolyte stock solution. An amount of 0.5 mL of the a-Amylase solution (1500 U/mL, made up in an SSF electrolyte stock solution), 25 μL of 0.3 M CaCl_2_ and 975 μL of ultrapure water were added. Afterwards, the mixture was incubated at 37 °C with shaking for 2 min.

Simulated gastric digestion: The oral digest was mixed with SGF at 1:1 (*v*/*v*) ratio containing 2000 U/mL pepsin and 0.15 mM CaCl_2_. The procedure was as follows: 10 mL of oral digest was mixed with 7.5 mL of the SGF electrolyte stock solution. An amount of 1.6 mL of pepsin solution (25,000 U/mL, made up in an SGF electrolyte stock solution), 5 μL of 0.3 M CaCl_2_, 200 μL of 1 M HCl and 695 μL of ultrapure water were added. The pH was adjusted to 3.0 with 1 M HCl. Afterwards, the mixture was incubated at 37 °C with shaking for 2 h.

Simulated intestinal digestion: The gastric digest was mixed with SIF at a 1:1 (*v*/*v*) ratio containing pancreatin (based on trypsin activity at 100 U/mL) and 0.24% (*w/v*) bile. The procedure was as follows: 20 mL of gastric digest was mixed with 11 mL of the SIF electrolyte stock solution. An amount of 5.0 mL of a pancreatin solution (800 U/mL based on trypsin activity, made up in an SGF electrolyte stock solution), 2.5 mL bile solution (38.4 mg/mL), 40 μL of 0.3 M CaCl_2_, 150 μL of 1 M NaOH and 1.31 mL of ultrapure water were added. The pH was adjusted to 7.0 with 1 M NaOH. Afterwards, the mixture was incubated at 37 °C with shaking for 2 h.

Simulated gastric or gastrointestinal digestion was immediately stopped by heating in a boiling water bath for 10 min [14]. After precipitation for soluble but undigestible protein fractions with trichloroacetic acid (5% (*v*/*v*), the final concentration at 4 °C for 1 h [25,26,27] and centrifugation (9000 rpm at 4 °C for 15 min), the supernatants (digestible fractions) were immediately stored at −80 °C until further analysis.

### 2.5. Crude Protein (CP) and Amino Acid (AA) Analysis

CP contents were determined following the Kjeldahl method (AOAC, 2010) using a Foss Tecator System (Höganäs, Sweden) with the conversion factor of 6.25 [28].

AA contents were determined in 5 mL of digestive liquid (0.2 g of flour) following hydrolysis in 5 mL of 6 N HCl containing 0.1% phenol for 24 h at 110 ± 1 °C in a hydrolysis tube [29]. After cooling and filtering, the hydrolysate was evaporated in a constant temperature water bath at 55–65 °C. The dry residue was diluted to 0.4 g/L (total AA concentrations) in 0.1 N HCl.

AA analysis was performed after pre-column derivatization using phenyl isothiocyanate as a derivatizing compound (Macklin Biochemical reagent, Shanghai, China). Chromatographic separation was achieved using an Agilent Advance Bio AAA column (4.6 × 100 mm, 2.7 μm) in a Shimadzu LC-15C system. Detection was carried out using a UV double wavelength detector (SPD-15C) with the absorbance wavelength of 254 nm, and the analysis time was 18 min. AA was separated using a gradient mobile phase consisting of Eluent A (10 mM Na_2_HPO_4_ and 10 mM Na_2_B_4_O_7_ in water; pH 8.2) and Eluent B (ACN/MeOH/H_2_O = 45:45:10). The gradient used was as follows: 0–0.35 min, 2%B; 0.35–6.9 min, 2%–22% B; 6.9–13.4 min, 22%–56% B; 13.4–13.5 min, 56%–100% B; 13.5–15.7 min, 100% B; 15.7–15.8 min, 100%–2% B and 15.8–18 min, 2% B. The flow rate was set at 1.0 mL/min, and the injection volume was 10.0 μL. Trp could not be identified due to its instability in acid hydrolysis conditions. Due to the same reason, Gln and Asn will be quantified in their acidic forms.

To assess the linearity of the calibration curve, 5 different concentrations (ranging from 0.125 to 2 μmol/μL) of an AA standard mixture were analyzed in triplicates through HPLC. The retention time of AA in test samples was similar to that of a standard AA mixture. The linear regression coefficient (R^2^) was higher than 0.99 for each standard AA in the concentration range of 0.125–2 μmol/μL, implying a good correlation between AA concentrations and peak areas within test limits (Table 1).

### 2.6. In Vitro Digestibility of Protein and Individual AA

The nitrogen content was determined by the Kjeldahl method (AOAC, 2010) using a Foss Tecator System (Höganäs, Sweden). Protein content was calculated using the conversion factors of 6.25 [28]. The *in vitro* protein digestibility was calculated as follows [30]:(1)$$In\;vitro\;\mathrm{protein}\;\mathrm{digestibility}\;\%=\;\frac{\left(\mathrm{Ns}-\mathrm{No}\right)\times 6.25\times V}{\mathrm{Nt}\times 6.25\times W}\times 100\%$$

Ns is %N in the supernatants after digestion; No is %N in blanks; V is volume of the supernatants after digestion; Nt is %N in cereal-based foods before digestion; and W is weight of cereal samples before digestion.

The *in vitro* AA digestibility was calculated by dividing the AA contents of the digestible fractions (supernatants) of the samples through that of the raw materials, by using the formula as follows:(2)$$In\;vitro\;\mathrm{AA}\;\mathrm{digestibility}\;\left(\%\right)=\frac{\mathrm{Fs}-\mathrm{Fo}}{\mathrm{Ft}}\times 100\%$$

Fs is individual AA content in the supernatants after digestion; Fo is individual AA content in blanks; and Ft is individual AA content in the cereals.

### 2.7. Digestible Indispensable AA Score (DIAAS) Calculation

The DIAA reference ratio of the cereal-based foods was calculated using the following equation [31]:(3)$$\mathrm{DIAA}\;\mathrm{reference}\;\mathrm{ratio}=\frac{\mathrm{mg}\;\mathrm{of}\;\mathrm{digestible}\;\mathrm{indispensable}\;\mathrm{AA}\;\mathrm{in}\;1\;g\;\mathrm{of}\;\mathrm{food}\;\mathrm{protein}\;}{\mathrm{mg}\;\mathrm{of}\;\mathrm{indispensable}\;\mathrm{AA}\;\mathrm{in}\;1\;g\;\mathrm{of}\;\mathrm{reference}\;\mathrm{protein}}$$
where the reference protein profiles were 2 age groups: children from 6 months to 3 years and older children, adolescents and adults [10].

The DIAAS values were also calculated for these two age groups using the following equation [31]:(4)DIAAS (%)=100× lowest value of DIAAR 

### 2.8. Statistical Analysis

Statistical analysis was performed using the IBM SPSS 26.0 software package. All results were given as means ± SD and were compared with one-way analysis of variance (ANOVA) with a 95% confidence interval.

## 3. Results

### 3.1. Comparisons of Crude Protein (CP) Contents and AA Compositions between Raw Grains and Protein Isolates of Three Cereals

The CP contents and AA contents in the raw grains and protein isolates of three cereals (millet, highland barley and buckwheat) are presented in Table 2. The CP contents of the three cereal grains were 11.7% for millet, 9.3% for highland barley and 12.1% for buckwheat, respectively. The CP contents of their protein isolates were 43.8%, 93.2% and 89.5%, respectively, and the protein isolates of millet had the lowest CP content. Due to the high hydrophobicity and intermolecular disulfide bonds of protein, traditional methods such as the isoelectric precipitation method might give limited results in the rate of protein extraction [32,33]. To overcome that, the enzymatic hydrolysis method was applied to millet protein extraction; however, the complex physicochemical interaction between protein and nonprotein components might contribute to the low protein content of protein isolates [34,35,36,37].

As presented in Appendix A, glutamic acid was the most abundant AA in the three cereal grains, with the content ranging from 18.3 to 23.7 g/100 g protein, whereas the contents of His, Met and Cys were at the minimum (below 2.9 g/100 g protein) in all the analyzed cereals, which were in accordance with that of their protein isolates.

However, the AA compositions of protein isolates were generally different from that of raw grains when all of them were expressed in a g per 100 g protein (Appendix A). For millet, the contents of Asp, Glu, Arg and Tyr in the protein isolates increased (*p* < 0.05), whereas the contents of Gly, His, Val, Met and Cys decreased (*p* < 0.05). Regarding highland barley, the protein isolates had the higher contents of Tyr, Phe, Glu and Pro (*p* < 0.05) and the lower contents of Met, Cys, Leu and Lys (*p* < 0.05). In buckwheat, the contents of Glu, Arg, Ile and Phe in the protein isolates increased (*p* < 0.05), whereas the contents of Asp, Ala, Met and Cys decreased (*p* < 0.05). The maximum variation was observed in the contents of sulfur AA (SAA, Met with Cys): a decrease in Cys (millet, 1.91% vs. 0.81%; highland barley, 2.64% vs. 1.04%; and buckwheat, 2.10% vs. 1.35%) and Met (millet, 2.65% vs. 1.75%; highland barley, 1.76% vs. 1.21%; and buckwheat, 2.31% vs. 1.15%) compared to raw grains (*p* < 0.01). Wang et al. [38] also found a decrease in the concentrations of SAA in the rice protein isolates, which might be attributed to oxidization during heat or alkaline treatments.

Although protein isolates had the higher contents of proteins and lower contents of impurities such as glucose, fatty acid and ash, the marked reduction in the contents of SAA may highly impact the AA analysis of cereals. Therefore, the AA analysis of uncooked cereals performed on the raw grains was more accurate than that performed on the protein isolates.

### 3.2. Effect of Processing Methods on Digestible AA Compositions of Three Cereals

Table 2 displays the CP contents and AA contents of raw grains and their digest from PG (DPG) and PF (DPF). The CP contents of the digest of cereal-based foods were generally different from the raw grains. The highest CP content was observed in the DPF of buckwheat (116.90 mg/g samples), whereas the DPG of millet had the lowest CP content (84.78 mg/g samples) among the digest of cereal-based foods.

Glu was the predominant AA in the digest of cereal-based foods (12.56–21.10 mg/g samples), whereas Cys had the lowest content (0.30–1.12 mg/g samples), which were in consistence with raw grains but for different quantities. Han et al. [39] also found Glu had the highest true ileal digestibility (TID) concentration, whereas Cys had the lowest TID concentration in the two varieties of millet and buckwheat in growing rats.

Figure 1 displays the comparative indispensable AA compositions of raw grains and their digest (DPG and DPF) based on g/100 g protein. The indispensable AA compositions of the digest of cereal-based foods were generally different from their raw grains. Overall, compared to raw grains, a significant decrease was observed in the contents of Cys and Ile in the digest of cereal-based foods (*p* < 0.01), whereas a marked increase was found in the content of Lys (*p* < 0.01). Meanwhile, the contents of some indispensable AAs, such as His, Thr and Val in the DPG samples were generally different from those in the DPF samples (*p* < 0.01), which meant the digestible AA compositions of cereals were significantly affected by processing.

### 3.3. Effect of Processing Methods on In Vitro Protein Digestibility of Three Cereals

The *in vitro* protein digestibility of cereal-based foods with raw flour as a control determined by the Kjeldahl method is given in Figure 2. At the gastric phase, the *in vitro* protein digestibility of millet was very low (12.6%) for raw flour; after processing, the digestibility remained unchanged for PG or increased for PF (*p* < 0.01). However, the digestibility of cereal-based foods produced by highland barley and buckwheat dropped significantly (*p* < 0.001), and the digestibility of PG was lower than that of PF (*p* < 0.01). Generally, the *in vitro* protein digestibility of cereal-based foods was below 20% at the gastric phase, which concurred with the reports by Gulati et al. [40] for cooked proso millet and Nunes et al. [41] for two cooked sorghum varieties, respectively.

At the intestinal phase, the *in vitro* protein digestibility of raw flour ranged from 37.9% (millet) to 70.0% (buckwheat), which was higher than that of cereal-based foods (*p* < 0.05). Generally, thermal processing decreased the digestibility, whereas the digestibility of PF was generally higher than that of PG (*p* < 0.01), which meant milling treatment improved digestibility. Milling treatment may break the inside insoluble fibrous cell wall structures, making protein more vulnerable to enzymatic attack [42,43]. The digestibility of highland barley-based foods and buckwheat-based foods ranged from 42.0% to 58.2%, whereas that of millet-based foods was quite low (PG, 26.4%; PF, 33.9%). According to Mertz et al. [44], the protein digestibility of cooked millet was also lower than that of some other cereals such as wheat or corn. This might be due to the formation of hydrophobic protein aggregates and a polyphenol–protein complex during heating in water [40,45].

### 3.4. Effect of Processing Methods on In Vitro AA Digestibility of Three Cereals

The *in vitro* digestibility of individual AA in cereal-based foods produced using grains (PG) or flour (PF) determined by HPLC is given in Table 3. The intestinal digestibility of total AAs (TAA) of PF was higher than that of PG, which showed a similar tendency with that based on total nitrogen. However, the digestibility of individual AA varied widely within a food. The digestibility of Cys (7.4–20.7%) and Ile (20.3–42.5%) was the lowest among all AAs, whereas the digestibility of Lys and Gly was relatively high (Lys, 56.4–84.5%; Gly, 65.8–80.8%). Han et al. [39] reported the TID of Lys was 96.0%, whereas that of Cys and Ile were 66.3% and 78.5%, respectively, on cooked grains from proso millet in growing rats. Similar results were found by Rafii et al. [9], who stated Lys was highly available (97%) in polished and cooked white rice for humans.

The digestibility of the individual AA in PG was less than or equal to that in PF, which meant milling treatment improved the digestibility of most AA. In millet, the digestibility of most AA in PG was lower than that in PF (*p* < 0.05) except for His, Thr, Val, Pro and Ala. Regarding highland barley, the digestibility of most AA in PG was lower than that in PF (*p* < 0.05) except for Ile, Asp, Arg and Gly. Likewise, for buckwheat the digestibility of most AA in PG was lower than that in PF (*p* < 0.05) except for Asp, Glu, Ile and Gly.

The digestibility of most AA (20.3%–47.2%) in millet was lower than highland barley (32.3%–87.5%) and buckwheat (33.9%–86.8%) (*p* < 0.05). However, the digestibility of Cys, Lys, Asp and Gly in PF of millet was the highest among all cereal-based foods. Han et al. [39] reported the TID of most AA in proso millet was lower than buckwheat, except that the TID of Lys in proso millet was higher than that in buckwheat.

### 3.5. Effect of Processing Methods on DIAAS of Three Cereals

Appendix A presents the AAS values of the indispensable AA in raw grains of three cereals. The raw grains of three cereals had unbalanced profiles, and the first limiting AA was Lys, except that buckwheat has no limiting AA for older children, adolescents and adults. For millet, the first limiting AA was Lys with 0.33 for children aged 6 months to 3 years and 0.39 for older children, adolescents and adults. Kalinova and Moudry [46] also found Lys as the first limiting AA with 0.47 for proso millet. Regrading highland barley, the first limiting AA was Lys with 0.56 for children and 0.67 for adults. Bai et al. [47] also found Lys (0.61) as the first limiting AA for highland barley. For buckwheat, the first limiting AA was Lys with 0.87 for children and none for adults. The high content in Lys for buckwheat (AAS, 0.96) was also reported by Motta et al. [48].

Table 4 shows the digestible indispensable amino acid (DIAA) reference ratio and the DIAAS of cereal-based foods. The results show that the DIAA reference ratios of indispensable AA dropped significantly, and most of them were far below 1after processing and digestion. The first limiting AA was Lys for foods made by millet and highland barley; however, the first limiting AA was Leu for buckwheat-based foods.

Despite the DIAAS values being different with a different reference pattern, the DIAAS values of PG were generally lower than those of PF in three cereals (*p* < 0.01), which meant the processing methods had a significant effect on their DIAAS values. The highest value was observed for highland barley for PG, whereas for PF the highest value was observed for buckwheat. The millet-based foods had the lowest values, either in PG or in PF. A similar finding has been reported by Han et al. [39], where they stated the DIAAS values of cooked grains from millet were the lowest among nine cooked cereal grains in growing rats. This might be because the digestibility of the protein and individual AA in millet was lower than that in highland barley and buckwheat.

## 4. Discussion

Cereals are the major source of energy in most diets consumed by humans and contribute to the protein requirements of humans, especially in developing countries [4,5]. Therefore, the determination of protein quality in cereal-based foods provides important information for formulating balanced diets. Generally, this includes a nutritional assessment performed on raw grains or protein isolates. In 2014, Cervantes-Pahm et al. [31] determined the DIAAS values of raw grains from eight cereals in growing pigs. In 2016, Mota et al. [49] reported the protein digestibility corrected amino acid score (PDCAAS) values of raw grains from pseudocereals (quinoa and amaranth). In 2018, Abelilla et al. [50] reported the PDCAAS values and DIAAS values of oat protein concentrate in growing pigs. However, cereals are usually processed using grains or flour before being consumed; further work is needed to estimate the impact of food preparation or processing on protein quality.

Ideally, the nutritional assessment of foods should be determined *in vivo* (in humans or animals); however, these invasive studies are costly and raise ethical issues. As a result, *in vitro* models have been developed to overcome these challenges and provide a useful tool for predicting protein quality in humans. The use of *in vitro* models was supported by the FAO/WHO [10].

In order to determine the DIAAS values of various foods, we utilized the INFOGEST static protocol [13], which involved simulated oral, gastric and intestinal digestion. After the digestion, multiple methods could be used for sampling treatment, including the inactivation of proteases and removal of soluble protein. To inhibit trypsin and chymotrypsin, AEBSF (4-(2-aminoethyl) benzenesulfonyl fluoride hydrochloride) was added as reported by Sousa et al. in 2023 [17]. Heat shock treatment has also been used in recent years to irreversibly inactivate proteases, when there was no consideration for the evaluation of biological activity [14]. Soluble protein was removed through various methods, such as precipitation with trichloroacetic acid, which has been commonly used in recent years [25,26,27]. In our study, we utilized heat shock treatment to inactivate proteases and precipitation with trichloroacetic acid to remove soluble protein, which were based on our analysis requirements for determining DIAAS values.

In this study, the DIAAS values obtained for cooked grains (PG) from millet and buckwheat were different from the results reported by Han et al. [39,51] in animal models. The DIAAS value obtained for cooked grains from millet was higher than that reported by Han et al. [51], who stated the DIAAS value was 19 for children in growing pigs. Furthermore, the DIAAS value obtained for cooked grains from buckwheat was lower than that reported by Han et al. [39] for children in growing rats. Nevertheless, the DIAAS values of cooked grains from buckwheat were lower than that from millet, which was consistent with the results reported by Han et al. [39] in animal models but for different values. Variation in the DIAAS values may be a result of the differences between the *in vitro* digestion model and pig model, the temperature of cooking, cereal varieties and the growing conditions of the cereals. In addition, the DIAAS value obtained for cooked grains from highland barley was 45, which was lower than the result reported by Cervantes-Pahm et al. [31], who stated the DIAAS value obtained for raw grains from dehulled barley was 51 in growing pigs. The decrease in the DIAAS value might be due to cooking, which damaged the digestibility of protein and individual AA.

According to the DIAAS cut-off value introduced by an FAO Expert Consultation report [10], only dehulled oats are considered as a good protein source for human consumption with a DIAAS value of 77. Han et al. [39] considered buckwheat and tartary buckwheat as potential good protein sources, because their DIAAS were 68 and 47 after they were cooked using grains, respectively. In this study, the DIAAS values of PF were 43 for highland barley and 52 for buckwheat. It is possible that highland barley and buckwheat are good protein sources when they are processed by milling before cooking.

## 5. Conclusions

This study evaluated the protein quality of cereal-based foods based on the digestible AA compositions. The AA compositions of protein isolates were generally different from that of raw grains; especially, the marked reduction in the contents of SAA may highly impact the AA analysis of cereals. Cooking treatment decreased the *in vitro* protein quality of all the investigated cereals. Regarding the processing methods, milling treatment generally improved the digestibility of protein and individual AA in foods. The intestinal digestibility of individual AA within a food varied widely, and the digestibility of Cys and Ile was the lowest among all AAs. Highland barley and buckwheat are the potential good protein sources when they are processed by milling before cooking, because of PF having high DIAAS values. The first limiting AA in processed foods was consistent with that in raw grains (Lys) for millet and highland barley, whereas the first limiting AA for buckwheat changed into Leu in processed foods. This study compared the amino acid compositions before and after grain processing and gastrointestinal digestion and highlight the impact of processing methods on the protein quality of three cereals. However, the inclusion of *in vitro* absorption models or animal experiments in future approaches may help elucidate if these findings are still valid in a more realistic scenario.

## Figures and Tables

**Figure 1 foods-12-01714-f001:**
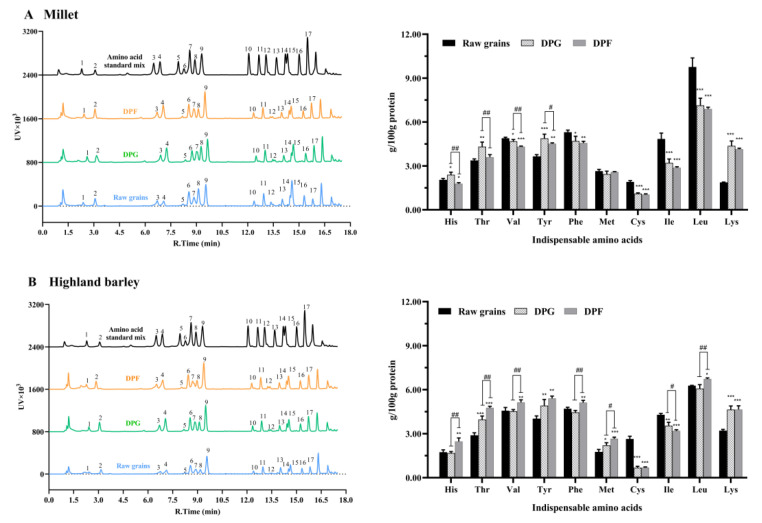
Amino acids composition of raw grains and digest from cereal-based foods from (**A**) millet, (**B**) highland barley and (**C**) buckwheat. **Left**: Chromatographs of 17 amino acids. This includes 1, Glu; 2, Asp; 3, Ser; 4, Gly; 5, His; 6, Thr; 7, Arg; 8, Ala; 9, Pro; 10, Tyr; 11, Val; 12, Met; 13, Cys; 14, Ile; 15, Leu; 16, Phe; and 17, Lys. **Right**: Indispensable amino acids content, g/100 g protein. DPG, simulated gastrointestinal digest from PG (cooked grains) and DPF, simulated gastrointestinal digest from PF (millet steamed buns, highland barley noodles and buckwheat noodles, respectively). Values of indispensable amino acid content were means ± SD (*n* = 3). * *p* < 0.05, ** *p* < 0.01, *** *p* < 0.001 comparison between DPG, DPF and raw grains, respectively; ^#^
*p* < 0.05, ^##^
*p* < 0.01, ^###^
*p* < 0.001 comparison between DPG and DPF.

**Figure 2 foods-12-01714-f002:**
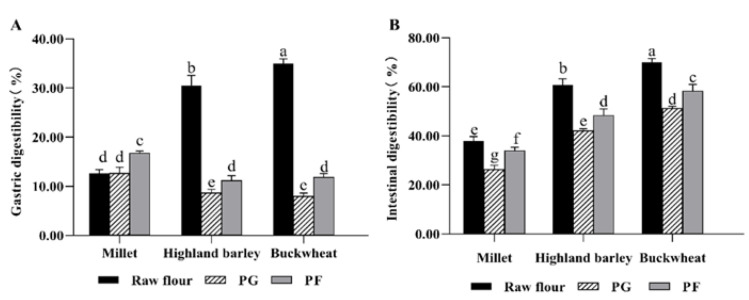
*In vitro* protein digestibility in (**A**) gastric and (**B**) intestinal phases of cereal-based foods produced by millet, highland barley and buckwheat. PG, cereal-based foods produced using grains (cooked grains); PF, cereal-based foods produced using flour (millet steamed buns, highland barley noodles and buckwheat noodles, respectively). Different letters on the top of the bars (mean ± SD, *n* = 3) represent significantly different (*p* < 0.05).

**Table 1 foods-12-01714-t001:** Calibration data of 17 amino acids (AA).

S.No	Amino Acid	Retention Time	Linear Regression (R^2^)	Calibration Curve
1	Asp	2.584	0.9993	y = 1,410,214.67x
2	Glu	3.371	0.9997	y = 1,626,141.02x
3	Ser	6.794	0.9999	y = 4,327,388.94x
4	Gly	7.166	0.9999	y = 4,420,685.42x
5	His	8.237	0.9998	y = 4,552,605.19x
6	Thr	8.567	0.9996	y = 4,531,925.27x
7	Arg	8.905	0.9994	y = 4,325,442.78x
8	Ala	9.204	0.9998	y = 4,710,639.21x
9	Pro	9.603	0.9999	y = 4,792,423.45x
10	Tyr	12.359	0.9999	y = 5,017,018.42x
11	Val	12.950	0.9996	y = 5,027,540.16x
12	Met	13.366	0.9997	y = 5,259,886.56x
13	Cys	13.976	0.9973	y = 3,948,177.56x
14	Ile	14.498	0.9989	y = 4,148,934.21x
15	Leu	14.619	0.9998	y = 5,319,782.40x
16	Phe	15.307	0.9999	y = 5,104,462.37x
17	Lys	15.792	0.9997	y = 8,961,547.43x

**Table 2 foods-12-01714-t002:** Crude protein contents (CP) and amino acid (AA) compositions of raw grains, protein isolates (PI) and digest from cereal-based foods produced by grains (DPG) or flours (DPF) from millet, highland barley and buckwheat (mg/g dry weight).

	Millet				Highland Barley				Buckwheat			
	Raw Grains	PI	DPG	DPF	Raw Grains	PI	DPG	DPF	Raw Grains	PI	DPG	DPF
CP ^1^	116.76 ± 1.88 ^a^	437.84 ± 3.32	84.78 ± 3.79 ^b^	107.74 ± 0.62 ^a^	92.89 ± 0.99 ^b^	932.14 ± 2.65	101.22 ± 5.51 ^a^	88.00 ± 2.90 ^c^	121.19 ± 0.13 ^a^	895.21 ± 1.38	100.25 ± 0.79 ^b^	116.90 ± 0.64 ^a^
Indispensable AA												
His	2.40 ± 0.09 ^a^	5.67 ± 0.17	2.06 ± 0.06 ^b^	1.93 ± 0.05 ^b^	1.62 ± 0.12 ^b^	16.81 ± 1.07	1.73 ± 0.03 ^b^	2.14 ± 0.11 ^a^	2.68 ± 0.15 ^c^	17.35 ± 1.77	2.18 ± 0.10 ^b^	3.10 ± 0.17 ^a^
Thr	3.94 ± 0.10 ^a^	13.27 ± 0.24	3.71 ± 0.13 ^a^	3.87 ± 0.15 ^a^	2.69 ± 0.13 ^b^	28.05 ± 0.85	4.08 ± 0.03 ^a^	4.15 ± 0.20 ^a^	5.64 ± 0.12 ^b^	41.55 ± 1.63	4.38 ± 0.22 ^c^	6.05 ± 0.15 ^a^
Tyr	4.27 ± 0.11 ^b^	18.95 ± 0.95	4.20 ± 0.08 ^b^	4.86 ± 0.07 ^a^	3.74 ± 0.14 ^b^	45.89 ± 0.64	5.05 ± 0.09 ^a^	4.71 ± 0.26 ^a^	4.16 ± 0.13 ^b^	33.74 ± 0.88	3.97 ± 0.25 ^b^	4.87 ± 0.10 ^a^
Val	5.71 ± 0.07 ^a^	20.20 ± 0.34	4.02 ± 0.09 ^c^	4.64 ± 0.02 ^b^	4.24 ± 0.16 ^a^	41.26 ± 0.41	4.67 ± 0.25 ^a^	4.48 ± 0.25 ^a^	5.86 ± 0.03 ^a^	41.58 ± 1.80	3.90 ± 0.02 ^c^	5.29 ± 0.31 ^b^
Met	3.09 ± 0.11 ^a^	7.64 ± 0.12	2.10 ± 0.07 ^c^	2.79 ± 0.03 ^b^	1.64 ± 0.12 ^b^	11.28 ± 0.51	2.27 ± 0.03 ^a^	2.31 ± 0.01 ^a^	2.80 ± 0.17 ^b^	10.33 ± 0.96	2.33 ± 0.04 ^c^	3.50 ± 0.13 ^a^
Cys	2.23 ± 0.09 ^a^	3.55 ± 0.44	0.96 ± 0.05 ^c^	1.12 ± 0.04 ^b^	2.46 ± 0.14 ^a^	9.66 ± 1.35	0.70 ± 0.05 ^b^	0.60 ± 0.03 ^b^	2.54 ± 0.20 ^a^	12.10 ± 0.93	0.30 ± 0.02 ^c^	0.61 ± 0.05 ^b^
Ile	5.66 ± 0.38 ^a^	20.64 ± 0.58	2.75 ± 0.07 ^b^	3.10 ± 0.07 ^b^	3.99 ± 0.06 ^a^	38.82 ± 0.78	3.64 ± 0.08 ^b^	2.78 ± 0.07 ^c^	5.26 ± 0.12 ^a^	42.42 ± 1.47	3.53 ± 0.12 ^b^	3.65 ± 0.05 ^b^
Leu	11.40 ± 0.59 ^a^	41.52 ± 1.63	6.13 ± 0.08 ^c^	7.41 ± 0.15 ^b^	5.83 ± 0.02 ^b^	55.17 ± 2.00	6.25 ± 0.11 ^a^	5.86 ± 0.21 ^b^	7.87 ± 0.12 ^a^	55.43 ± 1.95	4.23 ± 0.13 ^c^	6.68 ± 0.19 ^b^
Phe	6.20 ± 0.13 ^a^	23.09 ± 1.02	4.04 ± 0.07 ^c^	4.91 ± 0.13 ^b^	4.37 ± 0.08 ^a^	46.55 ± 0.58	4.59 ± 0.19 ^a^	4.46 ± 0.03 ^a^	5.41 ± 0.18 ^b^	44.83 ± 0.74	4.32 ± 0.05 ^c^	5.85 ± 0.21 ^a^
Lys	2.19 ± 0.02 ^c^	6.86 ± 0.18	3.75 ± 0.07 ^b^	4.46 ± 0.03 ^a^	2.98 ± 0.06 ^c^	24.13 ± 0.38	4.78 ± 0.06 ^a^	4.04 ± 0.06 ^b^	6.01 ± 0.07 ^b^	42.99 ± 0.74	5.37 ± 0.03 ^c^	7.15 ± 0.21 ^a^
Total ^2^	47.10 ± 0.93 ^a^	161.39 ± 2.07	33.73 ± 0.22 ^c^	39.10 ± 0.45 ^b^	33.55 ± 0.86 ^c^	317.62 ± 1.81	37.77 ± 0.55 ^a^	35.54 ± 0.84 ^b^	48.23 ± 0.87 ^a^	342.32 ± 5.56	34.51 ± 0.25 ^c^	46.74 ± 0.51 ^b^
Dispensable AA												
Asp	7.64 ± 0.30 ^b^	32.89 ± 0.69	7.40 ± 0.19 ^b^	15.72 ± 0.25 ^a^	6.32 ± 0.27 ^b^	66.22 ± 1.03	8.28 ± 0.39 ^a^	5.67 ± 0.29 ^b^	14.10 ± 0.10 ^a^	85.03 ± 3.58	10.97 ± 0.90 ^b^	10.23 ± 0.63 ^b^
Glu	21.31 ± 1.12 ^a^	87.74 ± 0.73	13.76 ± 0.35 ^c^	16.64 ± 0.62 ^b^	22.02 ± 0.55 ^a^	238.62 ± 6.74	21.10 ± 1.22 ^a^	12.56 ± 0.56 ^b^	23.08 ± 0.39 ^a^	190.63 ± 7.25	20.69 ± 1.38 ^ab^	20.27 ± 0.99 ^b^
Ser	5.17 ± 0.11 ^a^	18.13 ± 0.69	4.29 ± 0.12 ^c^	4.85 ± 0.12 ^b^	3.50 ± 0.14 ^c^	36.56 ± 0.16	5.26 ± 0.15 ^a^	4.16 ± 0.15 ^b^	5.79 ± 0.31 ^b^	43.47 ± 1.44	5.35 ± 0.26 ^b^	6.86 ± 0.10 ^a^
Gly	2.78 ± 0.07 ^c^	8.85 ± 0.23	4.74 ± 0.06 ^b^	5.43 ± 0.05 ^a^	3.09 ± 0.13 ^c^	29.48 ± 0.39	5.81 ± 0.28 ^a^	4.10 ± 0.08 ^b^	6.16 ± 0.20 ^a^	40.96 ± 2.24	6.42 ± 0.16 ^a^	6.67 ± 0.45 ^a^
Arg	6.57 ± 0.28 ^a^	29.87 ± 0.28	4.56 ± 0.28 ^c^	5.64 ± 0.05 ^b^	5.31 ± 0.53 ^ab^	56.69 ± 2.61	6.19 ± 0.30 ^a^	4.79 ± 0.26 ^b^	7.91 ± 0.08 ^b^	71.17 ± 2.86	7.27 ± 0.17 ^c^	8.47 ± 0.12 ^a^
Ala	8.83 ± 0.09 ^a^	32.59 ± 0.42	5.19 ± 0.25 ^c^	6.33 ± 0.11 ^b^	3.14 ± 0.12 ^c^	30.80 ± 0.70	4.51 ± 0.09 ^a^	3.68 ± 0.18 ^b^	4.88 ± 0.08 ^a^	30.22 ± 1.48	3.51 ± 0.11 ^b^	4.30 ± 0.46 ^a^
Pro	7.63 ± 0.04 ^a^	30.20 ± 0.21	5.51 ± 0.06 ^c^	6.26 ± 0.05 ^b^	8.44 ± 0.41 ^b^	94.97 ± 0.99	9.43 ± 0.30 ^a^	9.09 ± 0.13 ^ab^	3.92 ± 0.37 ^c^	32.65 ± 1.06	4.61 ± 0.12 ^b^	5.58 ± 0.26 ^a^
Total ^3^	59.94 ± 0.96 ^a^	240.27 ± 2.19	45.45 ± 0.51 ^b^	60.87 ± 0.60 ^a^	51.83 ± 0.08 ^b^	553.33 ± 8.62	60.59 ± 2.53 ^a^	44.06 ± 1.03 ^c^	65.83 ± 0.16 ^a^	494.12 ± 12.60	58.81 ± 2.49 ^b^	62.38 ± 2.41 ^ab^
TAA ^4^	107.03 ± 0.32 ^a^	401.66 ± 3.34	79.18 ± 0.71 ^c^	99.97 ± 0.72 ^b^	85.37 ± 0.92 ^b^	870.95 ± 8.62	98.37 ± 2.77 ^a^	79.60 ± 1.80 ^c^	114.06 ± 0.98 ^a^	836.44 ± 14.27	93.32 ± 2.28 ^c^	109.12 ± 2.29 ^b^

^1^ CP was based on an N-to-protein conversion factor of 6.25. ^2^ Total, combined total of indispensable AAs, including histidine, threonine, tyrosine, valine, methionine, cysteine, isoleucine, leucine, phenylalanine and lysine. ^3^ Total, combined total of dispensable AAs, including aspartic acid, glutamic acid, serine, glycine, arginine, alanine and proline. ^4^ TAA, combined total of AAs. ^a–c^ AA within a row in each kind of cereal with different superscript letters was significantly different (*p* < 0.05). Values of AA were means ± SD (*n* = 3); values of CP were means ± SD (*n* = 2).

**Table 3 foods-12-01714-t003:** Intestinal digestibility (%) of amino acid (AA) in cereal-based foods produced by grains (PG) or flour (PF) from millet, highland barley and buckwheat.

AA	Millet		Highland Barley		Buckwheat	
	PG	PF	PG	PF	PG	PF
Indispensable AA						
His	35.83 ± 0.95 ^c^	33.25 ± 0.81 ^c^	44.17 ± 1.09 ^b^	75.25 ± 4.62 ^a^	51.44 ± 2.57 ^b^	70.42 ± 5.62 ^a^
Thr	39.28 ± 1.34 ^d^	40.68 ± 1.62 ^d^	62.65 ± 1.24 ^b^	87.51 ± 3.19 ^a^	49.11 ± 2.19 ^c^	65.40 ± 0.85 ^b^
Tyr	41.02 ± 0.74 ^d^	47.19 ± 0.86 ^c^	55.65 ± 2.16 ^b^	71.27 ± 2.98 ^a^	60.30 ± 4.10 ^b^	71.43 ± 3.13 ^a^
Val	29.41 ± 0.72 ^c^	33.67 ± 0.20 ^c^	45.33 ± 1.56 ^b^	59.72 ± 2.56 ^a^	42.03 ± 0.37 ^b^	55.11 ± 4.83 ^a^
Met	28.37 ± 0.89 ^f^	37.44 ± 0.41 ^e^	57.15 ± 1.91 ^c^	80.05 ± 1.16 ^a^	52.52 ± 0.77 ^d^	76.01 ± 1.65 ^b^
Cys	17.93 ± 0.87 ^b^	20.77 ± 0.73 ^a^	11.80 ± 1.04 ^d^	13.88 ± 0.75 ^c^	7.44 ± 0.55 ^e^	14.56 ± 0.75 ^c^
Ile	20.31 ± 0.49 ^d^	22.70 ± 0.59 ^c^	37.69 ± 1.08 ^b^	39.49 ± 0.65 ^b^	42.46 ± 1.60 ^a^	42.35 ± 1.44 ^a^
Leu	22.44 ± 0.25 ^f^	26.95 ± 0.63 ^e^	44.23 ± 0.27 ^c^	56.98 ± 1.33 ^a^	33.94 ± 1.01 ^d^	51.72 ± 1.55 ^b^
Phe	27.19 ± 0.46 ^f^	32.83 ± 0.95 ^e^	43.30 ± 1.03 ^d^	57.84 ± 0.29 ^b^	50.43 ± 0.41 ^c^	65.96 ± 4.38 ^a^
Lys	71.54 ± 1.30 ^c^	84.46 ± 0.48 ^a^	66.16 ± 1.02 ^d^	77.04 ± 2.08 ^b^	56.43 ± 0.49 ^e^	72.57 ± 4.31 ^bc^
Total ^1^	29.89 ± 0.14 ^d^	34.41 ± 0.50 ^c^	46.42 ± 0.58 ^b^	60.03 ± 0.70 ^a^	45.21 ± 0.58 ^b^	59.10 ± 2.46 ^a^
Dispensable AA						
Asp	40.39 ± 1.01 ^d^	85.23 ± 1.65 ^a^	54.03 ± 1.35 ^b^	50.82 ± 1.98 ^b^	49.16 ± 3.81 ^bc^	45.08 ± 3.74 ^cd^
Glu	26.96 ± 0.65 ^d^	32.36 ± 1.21 ^c^	39.45 ± 1.49 ^b^	32.30 ± 1.08 ^c^	56.61 ± 3.50 ^a^	53.47 ± 1.27 ^a^
Ser	34.63 ± 0.92 ^e^	38.83 ± 1.01 ^d^	61.90 ± 0.59 ^c^	67.34 ± 2.56 ^b^	58.30 ± 2.54 ^c^	72.16 ± 2.54 ^a^
Gly	71.05 ± 1.01 ^cd^	80.82 ± 0.81 ^a^	77.48 ± 2.19 ^ab^	75.20 ± 1.46 ^bc^	65.83 ± 1.66 ^e^	65.97 ± 4.69 ^de^
Arg	28.98 ± 1.80 ^e^	35.58 ± 0.22 ^d^	48.08 ± 2.98 ^c^	51.05 ± 2.37 ^c^	58.07 ± 1.56 ^b^	65.25 ± 1.39 ^a^
Ala	24.53 ± 1.15 ^d^	29.73 ± 0.60 ^d^	59.27 ± 1.25 ^b^	66.40 ± 4.04 ^a^	45.54 ± 1.69 ^c^	53.72 ± 5.53 ^b^
Pro	30.17 ± 0.31 ^e^	34.00 ± 0.21 ^e^	46.05 ± 1.28 ^d^	61.05 ± 0.61 ^c^	74.34 ± 1.77 ^b^	86.77 ± 3.99 ^a^
Total ^2^	31.65 ± 0.29 ^d^	42.09 ± 0.51 ^c^	48.17 ± 0.92 ^b^	48.15 ± 0.61 ^b^	56.44 ± 2.12 ^a^	57.95 ± 2.29 ^a^
TAA ^3^	30.88 ± 0.22 ^e^	38.71 ± 0.41 ^d^	47.48 ± 0.61 ^c^	52.82 ± 0.54 ^b^	51.69 ± 1.03 ^b^	58.44 ± 1.99 ^a^

^1^ Total, the intestinal digestibility of total indispensable AAs, calculated by dividing the contents of total indispensable AAs of the digestible fraction of the samples through that of the raw materials. ^2^ Total, the intestinal digestibility of total dispensable AAs, calculated by dividing the contents of total dispensable AAs of the digestible fraction of the samples through that of the raw materials. ^3^ TAA, the intestinal digestibility of total AAs, calculated by dividing the contents of total AAs of the digestible fraction of the samples through that of the raw materials. ^a–f^ intestinal digestibility within a row with different superscript letters was significantly different (*p* < 0.05). Values were means ± SD (*n* = 3).

**Table 4 foods-12-01714-t004:** Digestible indispensable amino acid (DIAA) reference ratio and DIAAS of cereal-based foods produced by grains (PG) or flour (PF) from millet, highland barley and buckwheat.

Sample	Millet		Highland Barley		Buckwheat		Reference Pattern (mg/g Protein)
	PG	PF	PG	PF	PG	PF	
DIAA reference ratio (child (6 months to 3 years))							
Ile	0.31	0.34	0.51	0.53	0.57	0.58	32
Leu	0.34	0.39	0.43	0.54	0.33	0.52	66
Lys	0.24	0.27	0.38	0.43	0.49	0.64	57
Thr	0.41	0.43	0.59	0.81	0.73	0.99	31
Val	0.34	0.38	0.49	0.63	0.47	0.63	43
His	0.38	0.34	0.39	0.65	0.57	0.79	20
Sulfur AA	0.41	0.50	0.50	0.66	0.50	0.77	27
Aromatic AA	0.58	0.65	0.84	1.07	0.83	1.05	52
DIAAS (%) ^1^	24 (Lys) ^f^	27 (Lys) ^e^	38 (Lys) ^c^	43 (Lys) ^b^	33 (Leu) ^d^	52 (Leu) ^a^	
DIAA reference ratio (older child, adolescent, adult)							
Ile	0.33	0.36	0.55	0.56	0.61	0.62	30
Leu	0.37	0.42	0.46	0.58	0.36	0.56	61
Lys	0.28	0.32	0.45	0.51	0.58	0.76	48
Thr	0.54	0.59	0.74	1.01	0.91	1.23	25
Val	0.37	0.40	0.53	0.68	0.50	0.67	40
His	0.47	0.42	0.49	0.81	0.71	0.99	16
Sulfur AA	0.48	0.59	0.58	0.77	0.59	0.91	23
Aromatic AA	0.73	0.83	1.06	1.36	1.05	1.33	41
DIAAS (%) ^1^	28 (Lys) ^f^	32 (Lys) ^e^	45 (Lys) ^c^	51 (Lys) ^b^	36 (Leu) ^d^	56 (Leu) ^a^	

^1^ First-limiting AA was in parentheses. All DIAAS (%) value >100 indicated that no AA was limiting. ^a–f^ DIAAS (%) within a row with different superscript letters was significantly different (*p* < 0.05). Values were means ± SD (*n* = 3).

## Data Availability

The datasets generated for this study are available on request to the corresponding author.

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
