# Peer review of "Impact of Processing Methods on the In Vitro Protein Digestibility and DIAAS of Various Foods Produced by Millet, Highland Barley and Buckwheat"

_foods, 2023, doi:10.3390/foods12081714_

Round 1
Reviewer 1 Report
The manuscript "Impact of processing methods on the in vitro protein digestibility and DIAAS of various foods produced by millet, highland barley and buckwheat" by Fu and co-authors aims to compare protein degradation in cereal products using a validated in vitro method. With the current recommendations of a shift towards higher consumption of plant proteins it is an interesting and highly relevant topic since more knowledge is needed on different factors that influence protein degradation.
The report is interesting, but the manuscript needs to be improved.
Comments on the material and methods: Some of the methods have very limited details. Methods should be described in sufficient detail that a person can follow what was done without having to go on a literature search.
Some questions:
1. Where enzyme activities and bile concentration measured prior to the digestion experiments?
2. Were the samples frozen after digestion and before analysis?
3. How were the samples collected after gastric digestion treated?
4. Normalization of protein content, was this made? When the amount of substrate (protein) varies a lot, as it does for grains and isolates in the current study, this may result in different hydrolysis degree by the protein degrading enzymes. To allow comparison of protein hydrolysis between different samples it is sometimes recommended to normalize the protein content of different samples.
5. Why was the conversion factor 5.83 selected for calculation of total protein? This factor has been used by others too but since the factor 6.25 also is commonly used it can be good to “motivate” the choice.
Reviewer 2 Report
In this manuscript the in vitro determination of protein digestibility and DIAAS of various foods from three types of cereals is presented. It is a very timely subject as the nutritional value of non-conventional protein sources is a demand in the current scientific context.
The authors claim the INFOGEST in vitro method for determining DIAAS as a validated method on different protein sources with pigs and human digests as comparison. This makes the reader expect that they are using this method. However, the method used for digestion stop and separation of digested products is different as the one provided in Sousa et al., Food Chem 2023. They boil the digestion product, precipitate with TCA 5% and apparently they only collect the supernatants for analysis. This is different from the validated work where there is a rise of the pH, and MeOH precipitation, and centrifugation, the samples being separated into supernatant (S) and pellet (P) and treated independently for analysis. Also the blank (cookie) in the cited method is here substituted by water. Therefore, the authors have used a totally different method to calculate DIAAS, non-validated, and therefore cannot declare that they are using the INFOGEST method, as they have done in the objective (lines 48-50) and discussion (lines 393-395). They could propose their method to calculate this index, but this should be clearly stated in a rewritten version.
Round 2
Reviewer 1 Report
The manuscript has been revised by the authors and the text has been improved. It can be accepted for publication.
Reviewer 2 Report
The authors do not consider Trp because they cannot determine its content. However, they should justify this as if it would be a limiting amino acid, especially after processing, this would not be determined.